# A Novel *SLPI* Splice Variant Confers Susceptibility to Otitis Media in Humans

**DOI:** 10.3390/ijms26041411

**Published:** 2025-02-07

**Authors:** Christina L. Elling, Allen F. Ryan, Talitha Karisse L. Yarza, Amama Ghaffar, Erasmo Gonzalo d. V. Llanes, Jennifer M. Kofonow, Maria Rina T. Reyes-Quintos, Saima Riazuddin, Charles E. Robertson, Ma. Leah C. Tantoco, Zubair M. Ahmed, Abner L. Chan, Daniel N. Frank, Charlotte M. Chiong, Regie Lyn P. Santos-Cortez

**Affiliations:** 1Department of Otolaryngology-Head and Neck Surgery, School of Medicine, University of Colorado Anschutz Medical Campus, Aurora, CO 80045, USA; christina.elling@cuanschutz.edu; 2Department of Otolaryngology, School of Medicine and Veterans Affairs Medical Center, University of California San Diego, La Jolla, CA 92093, USA; afryan@health.ucsd.edu; 3Philippine National Ear Institute, National Institutes of Health, University of the Philippines Manila, Manila 1000, Philippines; tlyarza@up.edu.ph (T.K.L.Y.); edllanes1@up.edu.ph (E.G.d.V.L.); mtreyesquintos@up.edu.ph (M.R.T.R.-Q.); mjtantoco@up.edu.ph (M.L.C.T.); alchan@up.edu.ph (A.L.C.); cmchiong@up.edu.ph (C.M.C.); 4Newborn Hearing Screening Reference Center, National Institutes of Health, University of the Philippines Manila, Manila 1000, Philippines; 5Department of Otorhinolaryngology-Head and Neck Surgery, School of Medicine, University of Maryland, Baltimore, MD 21201, USA; aghaffar@som.umaryland.edu (A.G.); sriazuddin@som.umaryland.edu (S.R.); zmahmed@som.umaryland.edu (Z.M.A.); 6Department of Otolaryngology-Head and Neck Surgery, University of the Philippines College of Medicine—Philippine General Hospital, Manila 1000, Philippines; 7Division of Infectious Diseases, Department of Medicine, School of Medicine, University of Colorado Anschutz Medical Campus, Aurora, CO 80045, USA; jennifer.kofonow@cuanschutz.edu (J.M.K.); charles.robertson@cuanschutz.edu (C.E.R.); daniel.frank@cuanschutz.edu (D.N.F.)

**Keywords:** otitis media, microbiome, middle ear, nasopharynx, RNA-sequencing, *SLPI*

## Abstract

Otitis media is the most frequently diagnosed disease and a leading cause of hearing loss in young children. However, genetic contributors to susceptibility and pathogen–host–environment interactions in otitis media remain to be identified. Such knowledge would help identify at-risk individuals and effectively monitor, diagnose, and treat patients with otitis media. Through exome and Sanger sequencing, we identified a rare, deleterious splice variant *SLPI* c.394+1G>T co-segregating with otitis media in a large pedigree, with a genome-wide significant maximum LOD score of 4.59. Alternative splicing of *SLPI* was observed in saliva RNA of variant carriers. In bulk mRNA-seq data from an independent cohort of children with otitis media, *SLPI* was co-expressed with genes involved in infection, immune response, inflammation, and epithelial cell organization. After inoculation of non-typeable *Haemophilus influenzae*, *Slpi* was upregulated in polymorphonuclear leukocytes and epithelial cells of mouse middle ears. Furthermore, in the human middle ear, *Haemophilus* was significantly enriched in non-carriers, whereas *Family-XI-Incertae-Sedis* and *Dialister* were significantly enriched in variant carriers. Given the role of SLPI in immune modulation and host defense in mucosal epithelia, our findings support the *SLPI* variant as modulating susceptibility to otitis media.

## 1. Introduction

Otitis media (OM), i.e., middle ear (ME) infection and inflammation, is a leading cause of hearing loss and the most frequently diagnosed disease in young children worldwide [1]. OM is often preceded by viral upper respiratory infections which then promote the migration of bacterial otopathogens from the nasopharynx (NP) to the ME. The risk of developing OM is influenced by many environmental factors, as well as host genetics [2]. Despite these findings, there are still genetic contributors to be identified, and there is an unclear understanding of how host genetic variants impact the microbiotas of the ME and NP, thus creating an environment that favors infection and inflammation. This knowledge is important for improving understanding of the genetic architecture, as well as the pathogen–host–environment interactions underlying OM pathophysiology.

In an indigenous Filipino population (*n*~250) in which non-syndromic OM has a prevalence of roughly 50%, we previously identified multiple variants potentially conferring OM risk in this population [3,4,5]. Carriage of each of these variants in this cohort is associated with OM independent of other environmental factors and with changes in the ME and/or NP microbiotas [3,4,5,6,7]. There are still unresolved individuals with acute and/or healed OM in a single family branch within the cohort pedigree who are not carriers of any of these variants. Here, we present a novel rare and damaging variant in the gene encoding secretory leukocyte peptidase inhibitor (*SLPI*) that resolves this branch of the pedigree, co-segregates with OM in the entire indigenous cohort, affects splicing, and shifts the middle ear and nasopharyngeal microbiotas. Additionally, RNA-seq data from an independent human cohort and from mouse middle ear tissues support the role of *SLPI* in immune and epithelial defense against OM.

## 2. Results

### 2.1. Identification of the SLPI Splice Variant from Exome Sequence Data

A splice variant in *SLPI* (NM_003064:c.394+1G>T) was identified from the exome sequence data from individual IPOM-144, who is a member of the unresolved pedigree branch. At the time of examination and sample collection, IPOM-144 presented with acute OM yet is not a carrier of previously identified pathogenic variants in *SPINK5*, *FUT2*, or *A2ML1* (Figure 1) [3,4,5]. Using exome data from individual IPOM-144, variant annotation identified the presence of 95 variants that were rare and predicted to be damaging by ≥1 bioinformatics tools, excluding variants previously identified within this cohort but not associated with otitis media [4]. Of these variants, 26 were present in a minimum of three of the nine previously sequenced exomes [3,4,5,6]. Follow-up Sanger sequencing and linkage analyses of four of these variants—*CEP97* c.1922C>G, *ENPEP* c.212C>A, *CCDC91* c.800A>T, and *SLPI* c.394+1G>T—resulted in deflated LOD scores for three variants but had significant results for the *SLPI* splice variant.

In the indigenous Filipino cohort, the *SLPI* c.394+1G>T variant had a minor allele frequency (MAF) of 0.39 (Figure 1). Two-point linkage analysis resulted in a genome-wide significant maximum LOD score of 4.59 (θ = 0.05) in an affecteds-only model assuming autosomal dominant inheritance with full penetrance and no phenocopies. Furthermore, this variant is absent in DNA samples from the Cebu Longitudinal Health and Nutrition Survey (CLHNS) cohort [8] that is representative of the general Filipino population. It is also absent in variant databases GenomeAsia 100K, All of Us and Greater Middle Eastern (GME) Variome [9,10,11]. In the most recent version of the gnomAD v4.1.0 database, the variant is present in one out of 62,490 alleles (one heterozygous female in 31,245 individuals) with MAF = 1.6 × 10^−5^ in the “Remaining” population; however, it is absent in nine other gnomAD cohorts that have a cumulative total of 775,603 multi-ethnic individuals [12].

*SLPI* c.394+1G>T was a prime candidate variant based on gene expression patterns. There was high RNA expression in ME mucosa from an OM cohort in Colorado (median TPM [transcripts per million] = 2863) [13]. In the Genotype-Tissue Expression (GTEx) database [14], *SLPI* RNA is predominantly expressed in the minor salivary gland (median TPM = 13,610), esophageal mucosa (median TPM = 1418), and lung (median TPM = 880). In the Human Protein Atlas [15], high protein levels are reported in the nasopharynx and cervix (which contains mucosal epithelia) and medium levels in tissues that include the bronchus. In the human Ensemble Cell Atlas (hECA) [16], *SLPI* RNA has the highest expressing ratio in the esophagus (70.6% expressing), pleura (44.2%), and bronchi (44.1%), as well as perineural epithelial cells (73.3%), mesothelial cells (50.1%), intestinal gland cells (48.3%), and Paneth cells (48.3%).

### 2.2. SLPI c.394+1G>T Results in Alternative Splicing and Exon Skipping

We sought to determine if the functional effects of the *SLPI* variant are due to alternative splicing and exon skipping. SpliceAI predicts a pathogenic splice donor loss (Δ score = 0.76) at the natural donor site of exon 3 [17]. Splicing of saliva RNA isolated from IPOM-51 (“*SLPI*-homozygous”) showed one expected size product. RNA splicing for IPOM-39 (“*SLPI*-heterozygous”) showed two products, one expected and one spliced, and thus accurately reflects their genotypes as compared to RNA from a wildtype individual not carrying the *SLPI* variant (Figure 2A,C). Sanger sequencing of cDNA generated from the saliva RNA confirms that exon 3 is skipped in IPOM-51, who is homozygous for the variant (Figure 2B). The skipping of exon 3 is predicted to remove 51 out of 132 amino acids that are located at the C-terminus. According to the InterPro database, the skipped gene region includes the elafin-like WAP-type four-disulfide core domain, which is the active site of the anti-protease activity of SLPI [18,19].

### 2.3. SLPI Is Co-Expressed with Genes Involved in Infection and Immune Responses

We had previously available bulk mRNA-seq data from an independent pediatric cohort with OM [6], which was used to further determine the effects of *SLPI* expression levels on the human transcriptome. Spearman’s correlation analysis of saliva identified 214 genes that are moderately and significantly (ρ_abs_ ≥ 0.5; false discovery rate or FDR-adjusted-*p* < 0.05) co-expressed with *SLPI*, of which 205 are positively correlated and 9 are negatively correlated (Appendix A). These include *ARSA*, *MIF*, and *SERPINE1*, which were previously identified as candidate genes for OM [20,21,22].

Differential expression analysis was also performed comparing high- and low-*SLPI* expressers in the same dataset. High- versus low-*SLPI* expression was defined by whether samples were above the median *SLPI* expression for the cohort (in which case they were “high”) or below the median (“low”). Nineteen differentially expressed genes were identified, of which seven genes were downregulated and twelve genes were upregulated in high expressers (Appendix A). Fifteen of these genes, along with *SLPI*, are connected by a single protein–protein interaction (PPI) network (Appendix A). Pathway enrichment analysis of this network using the KEGG database revealed 106 significant pathways (FDR-adjusted-*p* < 0.05), including many associated with infection, immune response, inflammation, and epithelial cell organization (Appendix A). Comparing the list of correlated and differentially expressed genes, five genes overlap and are connected through the PPI network, namely *AKR1C1*, *HNRNPD*, *MT-ATP8*, *SURF4*, and *UGDH* (Appendix A).

### 2.4. Increased Slpi Expression in Acutely Infected Mouse Middle Ears

To assess which cell types in the ME express *Slpi* under homeostatic conditions versus acute OM from non-typeable *Haemophilus influenzae* (NTHi) infection, we performed single-cell RNA-seq on ME tissues from a mouse model. In control mice, *Slpi* had mild-to-moderate expression in a small proportion of epithelial cells including basal epithelial cells. There was also strong expression in an estimated 10% of lymphocytes and some monocytes (Figure 3A). Prior to inoculation, the total number of cells expressing detectable *Slpi* mRNA was 129 out of 2818 (4.6%). After NTHi inoculation, a marked increase in *Slpi* expression occured across all cell types, with polymorphonuclear and epithelial cells demonstrating strong expression in the majority of each cell population (Figure 3B). The number of cells expressing detectable *Slpi* mRNA at 6 h after inoculation was 2041 out of 2557 (79.8%) cells in the sample. In the control sample, the highest level of expression was in epithelial cells, but the log_2_ median normalized expression was zero. In the 6-hour sample, the median normalized expression was 3.32-log_2_. As a control, we assessed the expression of beta-actin in the epithelial cells of each sample. Median *Actb* mRNA-normalized expression in control epithelial cells was 3.24-log_2_, and for the infected sample was 3.85-log_2_.

### 2.5. Microbiota Shifts in the Middle Ear and Nasopharynx According to Carriage of the SLPI Variant 

To determine whether the *SLPI* splice variant is associated with changes in the ME and NP microbiotas, we collected ME and NP swabs and performed microbiota sequencing and analyses. In the ME, biodiversity as measured by both alpha- and beta-diversity indices were not significant when comparing *SLPI* c.394+1G>T variant carriers (*n* = 7) to non-carriers (*n* = 9) (Figure 4A, Appendix A). On the other hand, *Dialister* (false discovery rate or FDR-adjusted-*p* = 8.38 × 10^−16^) and *Clostridiales Family-XI-Incertae-Sedis* (FDR-adjusted-*p* = 8.38 × 10^−16^) were enriched in *SLPI* variant carriers, (Figure 4B). With NTHi being the most commonly identified bacterium in OM [23], as expected, *Haemophilus* (FDR-adjusted-*p* = 2.97 × 10^−6^) was significantly enriched in wildtype compared to variant carriers (Figure 4B).

It is not unusual to have a low sample size for ME microbiota analyses because these samples are difficult to obtain via surgery and often have low yields of nucleic acids. Therefore, the nasopharynx is often used as a proxy, particularly because it can serve as a reservoir for potential otopathogens. In the NP, alpha-diversity indices between *SLPI* variant carriers (*n* = 20) and non-carriers (*n* = 44) were not significant (Appendix A). On the other hand, beta-diversity differed between carriers and non-carriers (nominal-*p* = 0.04) after adjusting for sequencing batch (*p* = 1 × 10^−6^) and age (*p* = 0.67; Appendix A). In *SLPI* variant non-carriers, the uncultured taxon WCHB1-69 (FDR-adjusted-*p* = 5.29 × 10^−20^) and *Lachnospiraceae* (FDR-adjusted-*p* = 0.001) were enriched as compared to *SLPI* variant carriers (Appendix A).

## 3. Discussion

Here, a novel splice variant in *SLPI* was identified in a proband with acute OM that is not a carrier of previously identified pathogenic variants in *A2ML1*, *SPINK5*, or *FUT2.* This variant co-segregates with OM in a large pedigree, resulting in a statistically significant LOD score of 4.59. We also examined the impact of this variant on the microbiotas of the ME and NP in this cohort and observed a significant change in beta-diversity in variant carriers, along with an increase in the abundance of multiple taxa. The role of *SLPI* c.394+1G>T in OM is further supported by: (a) an RNA splice variant assay according to the *SLPI* variant genotypes; (b) differential expression and pathway analyses using human mRNA-seq data; and (c) scRNA-seq data from mouse ME before and after NTHi inoculation.

Given the results of the splice variant analysis of saliva RNA, in which it is evident that exon 3 is skipped, the functionality of the SLPI protein likely is negatively impacted not only in the oral cavity but in other mucosal epithelia tissue as well. It is interesting to note that in the *SLPI*-heterozygous individual, the two alleles are not equally represented, but rather there seems to be a greater abundance of the variant allele compared to the wildtype allele (Figure 2). There are a few possible explanations for this observation: (a) variant RNA transcripts are undergoing nonsense-mediated decay, a mechanism by which cells surveil mRNA transcripts in the cytoplasm and trigger degradation of aberrant transcripts that contain premature termination codons, although this seems unlikely given the greater abundance of the variant transcript [24]; (b) random monoallelic gene expression, such as through imprinting or as the result of stochastic low-probability expression in cells that can then be selected for clonal expansion [25,26,27]; or (c) as a result of other *cis-*acting genetic variants [28]. More in-depth research of transcriptional regulation of *SLPI*, given the multiple variants identified in the indigenous population, is needed to further understand this difference in allele expression but is beyond the scope of this study.

The SLPI protein is ubiquitously present in mucosal epithelial tissues and is an important immune modulator, as well as part of host defense against serine proteases [29]. Macrophages and neutrophils are among the cell types producing SLPI. SLPI has been shown to inhibit neutrophil extracellular trap (NET) formation. NETs include enzymes such as neutrophil elastase, a serine protease, and play a crucial role in host defense against pathogens but can also contribute to inflammatory disease and autoimmunity if left unchecked [30,31]. The single-cell RNA-seq data from infected mouse ME presented here (Figure 3B) supports this role of *Slpi* in OM. At 6 h post-infection, *Slpi* is moderately to strongly expressed in nearly all monocytes (which are precursors to macrophages) and polymorphonuclear leukocytes (which neutrophils are a subset of), as well as epithelial cells. The dramatic infection-related increase in the number of cells expressing *Slpi* mRNA and in their expression levels, combined with the equivalent scRNA-seq metrics for the two samples (Appendix A), indicate that the difference is not the result of technical issues.

Interestingly, we have now identified rare variants in three protease inhibitor genes (*A2ML1*, *SPINK5*, and *SLPI*) in the same indigenous population, suggesting that this class of proteins may also have a beneficial effect given their environment, potentially in other organ systems. That variants modulating susceptibility to OM have other protective roles in various diseases or states is not unusual, e.g., *FUT2* p.Trp154* and p.Arg202* [5]. Note also that each protease inhibitor encoded by these genes has a different set of protease targets and different localization patterns within the ME, suggesting that each protease inhibitor has a unique role in the ME [3,4].

SLPI is an effective antimicrobial agent and in sinonasal mucosal secretions works against both Gram-negative and Gram-positive bacteria. Among the bacteria regulated by SLPI are *Pseudomonas aeruginosa* and *Staphylococcus aureus*, which are two well-known otopathogens [32,33]. Furthermore, human bronchial epithelial cells demonstrate a marked increase in SLPI levels after exposure to *P. aeruginosa* in vitro [34]. Although neither of these taxa were significantly different in relative abundance by variant carriage in the results presented here, they play important roles in OM and the effect of SLPI and its variants may still have a role not yet elucidated, such as altering commensal taxa levels and/or enabling further colonization by multiple pathogens. Additionally, SLPI has been shown to be protective against viral infections such as by HIV-1 or human papillomavirus, including in other mucosal sites [35,36,37,38,39]. Unfortunately, we only had limited viral data from indigenous Filipino children, which was not associated with any of the identified genetic variants. Given the remote location of the indigenous community, it was difficult to obtain sufficient longitudinal data that would determine the association between the *SLPI* variant and otopathogenic viruses, most of which can only be detected in the mucosal tissues transiently [40].

*Clostridiales Family-IX-Incertae-Sedis* has been identified in the NP microbiotas of children with acute OM, as well as the nasal cavity of an individual with chronic rhinosinusitis, a chronic inflammatory disease also related to bacterial infection [40,41]. *Dialister* has been previously detected in nasal and oral mucosal secretions, as well as a cholesteatoma sample, but has otherwise not previously been identified with significance in OM [42,43,44,45,46]. It is possible that *Dialister* is part of the normal oral and nasal microbiota and an opportunistic pathogen of the ME. *Lachnospiraceae* has also been identified as enriched in the NPs of children with recurrent respiratory tract infections and in cholesteatoma, so it is not unlikely that it is involved in OM dysbiosis and has a potentially pathogenic role [47,48]. *WCHB1-69* is a relatively understudied taxon in the context of human microbiotas and disease, but it has been identified in aquatic environments. It is likely that some pathogens involved in OM pathogenesis in this community were contracted from swimming in contaminated seawater [4], which may have been a source of *WCHB1-69.* Note that adequate negative controls were used in this study, and it is unlikely that these identified taxa were due to selective sample contamination. Furthermore, sample size for the ME microbiota data is limited due in part to the samples themselves having a low abundance of bacterial DNA by nature, and, despite having collected more samples, a proportion of the samples often do not pass QC thresholds for read counts or quality of reads. Because these are community-based samples, we were not able to collect ME samples from individuals not actively experiencing OM, such as those indicated as “normal” or “healed” without eardrum perforations (Figure 1).

Altogether, the data presented here suggest that the *SLPI* splice variant is involved in the pathology of OM in this cohort. Given the role of *SLPI* in immunomodulation as both part of host defense against pathogens, as well as mitigating inflammatory response through the protection of epithelial cells from endogenous proteases, we hypothesize that this variant is contributing to OM risk through its (A) negative impact on the ability of *SLPI* to maintain host defense and (B) impaired ability to protect the host epithelial tissues against chronic inflammation and associated damage during innate immune response to infection. Unfortunately, this variant has not been replicated in any of the exome data from additional cohorts we have data for, and no other predicted-to-be-damaging variants in *SLPI*, whether common or rare, have been identified in these cohorts either [3,4,5,6]. We conclude that *SLPI* is a strong candidate gene for susceptibility to OM in humans and is a potential target for future therapies against OM.

## 4. Materials and Methods

### 4.1. Ethical Approval

Human studies were approved by these institutional review boards (IRB): Colorado Multiple IRB; National Commission on Indigenous Peoples; and the University of the Philippines Manila. Informed consent was obtained from all study participants. Mouse studies were approved by the Institutional Animal Care and Use Committee (IACUC) of the Veterans Affairs Medical Center, San Diego.

### 4.2. DNA Isolation and Sequencing

The indigenous community was visited seven times to recruit pedigree members for study. All individuals with any OM type (acute, recurrent, chronic, effusive, and/or healed) were considered affected. The OM diagnosis was based on the last examination available for each individual. A total of 152 indigenous Filipinos provided clinical data and saliva samples for DNA isolation using Oragene kits (DNAgenotek, Ottawa, ON, Canada).

Three DNA samples from indigenous Filipinos with OM but negative for *A2ML1*, *FUT2*, and *SPINK5* variants were submitted for exome sequencing at the Northwest Genomics Center as a paid-for core service. Sequence capture was performed using the Roche NimbleGen SeqCap EZ Human Exome v.2.0 Library (Pleasanton, CA, USA). Exome sequencing was performed (average depth ~60×) using an Illumina HiSeq (San Diego, CA, USA). Burrows–Wheeler Aligner and the Genome Analysis Toolkit (GATK) were used to generate BAM and VCF files, respectively [49,50]. The *SLPI* variant was Sanger-sequenced in all available indigenous Filipino DNA samples.

To estimate the Filipino MAF of the identified variants, DNA samples from unrelated Filipinos (*n* ≥ 88) were obtained from the CLHNS cohort and Sanger-sequenced for these variants. The CLHNS DNA samples are from a community-based birth cohort recruited in 1983–1984 in order to study health and nutrition outcomes, but not OM [8]. The MAF of the *SLPI* variant was also checked in the GenomeAsia 100K (Southeast Asia) database, which includes 52 DNA samples from Negrito groups in the Philippines [9]. Additional databases checked were gnomAD v4.1.0, GME Variome, and All of Us [10,11,12].

### 4.3. Bioinformatic and Linkage Analyses

Rare variants from the exome data were annotated as previously described [3,4,5,6]. ANNOVAR software was used to annotate variants using the hg19 reference sequence, gnomAD, avsnp150, and dbNSFP33a databases [51,52]. Variants identified in the exome data were selected for further analysis if: (a) they passed all GATK quality control filters; (b) they were absent in gnomAD (given the uniqueness of this cohort and their lack of representation in gnomAD); (c) for indels, if they were deemed disease-causing by MutationTaster [53]; and (d) for single-nucleotide variants, they had a scaled Combined Annotation Dependent Depletion (CADD) score ≥20 [54] plus predicted damaging by ≥1 bioinformatics tools from dbNSFP. Furthermore, variants were filtered and prioritized based on their presence in other sequenced exomes from this cohort, as well as GTEx tissue expression and Human Protein Atlas expression [14,15].

Among the indigenous Filipinos, 136 ascertained individuals were connected by a single pedigree (Figure 1). Initially the pedigree was split into four branches in order to be accommodated by the software MERLIN to check for potential genotype inconsistencies [55], after which for three individuals the genotypes were flagged and changed to unknown. Two-point affecteds-only analysis was performed using Superlink software [56] to determine linkage between OM and the *SLPI* variant wherein if a wildtype affected individual carries any of the pathogenic *A2ML1*, *SPINK5*, or *FUT2* variants, the specific individual was considered a phenocopy. The whole pedigree was used as input in Superlink, which included a built-in check for Mendelian inconsistencies. A variant MAF of 0.000001 was used given the absence or rarity of the variant in multiple public databases and the CLHNS cohort. Parameters included autosomal dominant inheritance with full penetrance and with disease MAF set to equal variant MAF.

### 4.4. RT-PCR Assay of SLPI Splicing and SpliceAI Analysis

RNA samples from the saliva of three indigenous Filipinos (1 wildtype, 1 heterozygous, 1 homozygous) were used to amplify cDNA using SMARTScribe Reverse Transcriptase (639538) enzyme (Takara Bio USA, San Jose, CA, USA) based on the manufacturer’s protocol. CloneAmp HiFi PCR Premix (639298, Takara Bio USA) was used to amplify gene-specific amplicons using primers 5′-CCTTCAAAGCTGGAGTCTGTCC-3′ (forward primer from exon 2) and 5′-TGCAAAGAGAAATAGGCTCGTTT-3′ (reverse primer from exon 4). PCR products were run on gel, and products were confirmed via Sanger sequencing.

SpliceAI is a bioinformatic tool that utilizes deep learning to predict the impact of variants on splicing based on sequence and was applied to the *SLPI* variant. Delta (Δ) scores for variants range from 0 to 1 and reflect the probability of disrupting splice, with a higher score indicating a greater likelihood of impacting splicing, therefore being more likely to be a pathogenic variant [17].

### 4.5. Differential Expression from Bulk mRNA-Sequence Data from Human Samples and Network and Pathway Analyses

Previously available bulk mRNA-seq data from 30 children with OM [6] were re-analyzed to determine the co-upregulation and differential expression patterns of other genes according to *SLPI* expression levels. In summary, 30 salivary RNA samples (median RIN = 7.1) were submitted for bulk mRNA-sequencing, as previously described [6]. RNA samples were processed using the Nugen Trio RNA-Seq Kit (Tecan, Redwood City, CA, USA). Sequencing was performed on an Illumina HiSeq 4000 with an average of 31 million reads per sample.

Spearman’s correlation analysis to determine co-expressed genes was performed in R. Results were considered significant for genes with Spearman’s ρ > 0.5 or < −0.5 and FDR-adjusted-*p* < 0.05. For differential expression analysis, samples were divided into low-expressers and high-expressers of *SLPI* using median *SLPI* expression as the threshold. Differential expression analysis was performed using the DESeq2 package in R, with correction for age, sex, and batch effects [57,58]. Results were considered significant for genes with log_2_-transformed fold change >±2 and FDR-adjusted-*p* < 0.05 using the Benjamini–Hochberg method. From post hoc analysis using the RnaSeqSampleSize app (https://cqs-vumc.shinyapps.io/rnaseqsamplesizeweb/, accessed 25 January 2025), given a sample size of *n* = 30, maximum dispersion of 0.01, and 12,000 genes for testing, our RNA-seq analysis has 98% power to detect differential expression according to *SLPI* expression.

The resulting nineteen significantly differentially expressed genes and *SLPI* were used as input into NetworkAnalyst for construction of a PPI network using the IMEx interactome database [59].

### 4.6. Slpi Expression in Mouse Middle Ear

All animal experiments were performed according to the recommendations of the Guide for the Care and Use of Laboratory Animals of the National Institutes of Health and carried out in strict accordance with an approved IACUC protocol (A13-022) of the Veterans Affairs Medical Center (San Diego, CA, USA). All animal experiments employed the best efforts for minimizing animal suffering under general anesthesia according to the NIH guidelines.

Single-cell samples for RNA-sequencing were generated from the entire contents of the mouse ME and *Slpi* using previously described methods [60]. For each of three independent samples, tissue was harvested from both ears of six young adult C57BL/6J mice 6 h after inoculation of the ME with NTHi. Single-cell libraries were generated using the 10X Genomics (Pleasanton, CA, USA) Chromium Single Cell 3′ Reagent Kit V2. cDNA synthesis, barcoding, and library preparation were then carried out on a 10X Genomics Chromium Controller. After validating the quality of the cDNA library, sequencing was performed on an Illumina HiSeq 2500. Reads were demultiplexed and aligned to the murine reference genome (mm10 with annotations from Ensembl, release 84). Then, 10X Genomics Cellranger aggr and Seurat were used to generate PCA clustering [60,61]. The expression of well-recognized marker genes identified 24 distinct cell types. Linearized relative expression levels of each gene examined in this study were log-transformed from single-cell mRNA copy numbers, normalized, and scaled for each cell type. Data were visualized in 10X Genomics cLoupe, with unique molecular identifiers expressed colorimetrically for each cell.

### 4.7. 16S rRNA Gene Sequencing and Microbiota Analyses

Outer ear, ME, nasopharyngeal (NP), and oral cavity swabs were obtained from 90 indigenous Filipinos with and without OM as previously described [4,5]. Briefly, samples were collected using Oragene P117 prototype kits, and microbial DNA was isolated using the Epicentre MasterPure™ Kit (LGC Biosearch Technologies/Lucigen, Middleton, WI, USA). Bacterial profiles were determined by broad-range PCR amplification and sequence analysis of the V1-V2 regions of the 16S rRNA gene via paired-end sequencing on Illumina MiSeq using the 600 cycle version 3 kit. After removing poor-quality and chimeric reads, merged sequences were aligned and classified with SINA (1.3.0-r23838) using the 418,497 bacterial sequences in Silva 115NR99 as reference configured to yield the Silva taxonomy [47,62,63,64,65]. Taxonomic assignment by SINA used the lowest common ancestor approach with default parameters. This process generated a median of 38,635 sequences/sample (interquartile range: 29,075–54,710). Goods coverage scores were ≥99% for all samples at the rarefaction point of 10,000 sequences, indicating adequate depth of sequence coverage for all samples.

Bacterial alpha-diversity indices (richness [Chao1], complexity [Shannon H], and evenness [Shannon H/Hmax]) were inferred through 1000 replicate re-samplings at a rarefaction point of 10,000 sequences using Explicet v2.10.8 [66]. Between-group differences in these indices were evaluated via Wilcoxon test. Associations of individual taxon abundances with the *SLPI* variant were assessed by DESeq2 [57], with sequencing batch as a covariate. To minimize multiple comparisons, only taxa with a prevalence >10% and relative abundance >1% were included in the analysis. Between-group differences in overall microbiota composition (beta-diversity) were assessed via permutational multivariate ANOVA (PERMANOVA), as implemented by the adonis2 function of the vegan R package 2.5-6 [67]. These tests used the Aitchison dissimilarity index and adjusted for age, sex, OM type, and batch effects. *p*-values were ascertained through one million label permutations. R 4.1.0 software was used for data analyses and figure generation.

## Figures and Tables

**Figure 1 ijms-26-01411-f001:**
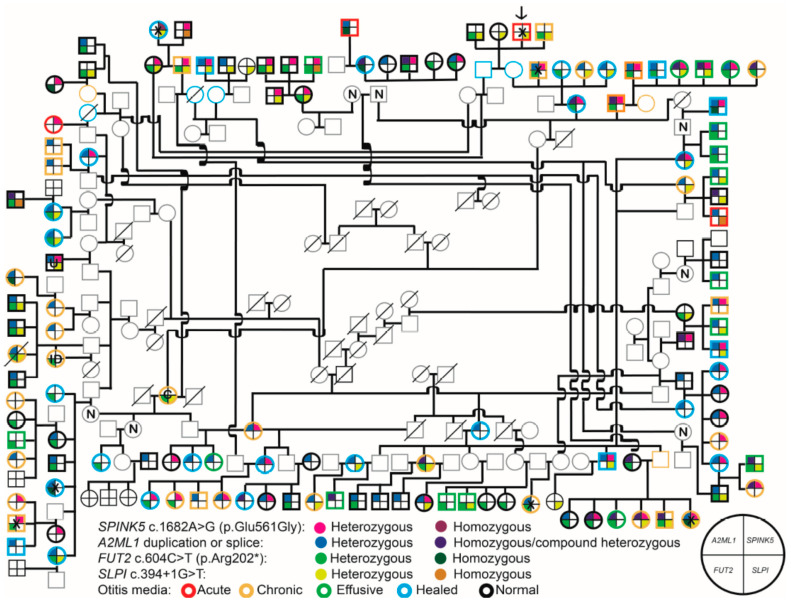
Indigenous Filipino pedigree showing OM status and genotypes. Pedigree demonstrating relationships of 136 individuals and their genotypes for pathogenic variants in *A2ML1*, *FUT2*, *SPINK5*, and *SLPI* variants. IPOM-144 is indicated by an *arrow* at the top of the pedigree and is a member of the previously unresolved branch. Two additional OM-affected children with new exome data cannot be connected to the pedigree and did not have novel variants, and are therefore considered phenocopies. Individuals who are wildtype for all the identified pathogenic variants are indicated by black crosshairs with no fill. Variant carriage is indicated by symbol quadrant and color according to the legend below the pedigree. Individuals with no crosshairs are those for whom we do not have genotype data. Otitis media type is indicated by the outline of each individual as indicated in the legend. All individuals with acute (*red*), chronic (*yellow*), effusive (*green*), and healed (*blue*) otitis media were considered affected. Individuals with a *black outline* have normal ears, and individuals with a *gray outline* are those for whom we lack phenotype data. X = DNA sample sent for exome sequencing; N = normal ears, no DNA sample; U = unknown phenotype; ID = intellectual disability; C = cholesteatoma. Adapted from Frank et al., 2021, this figure represents our most complete record of pedigree relations to date and was noted in previous publications for *SPINK5*, *FUT2*, and *A2ML1* [3,4,5,6].

**Figure 2 ijms-26-01411-f002:**
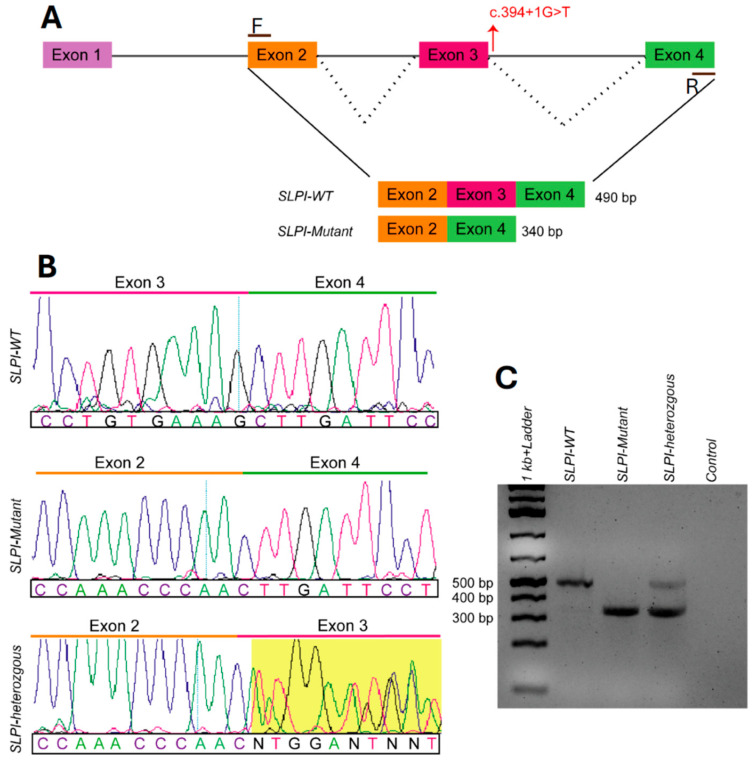
Splicing assay of saliva RNA samples. (**A**) The *top panel* demonstrates the splicing isoforms wherein transcripts in the variant lack exon 3. (**B**) The *bottom left panels* demonstrate sequencing results for each genotype. (**C**) The *bottom right panel* demonstrates the genotype differences in band sizes corresponding to the exon being skipped in homozygous individuals and the presence of both alleles in heterozygous individuals.

**Figure 3 ijms-26-01411-f003:**
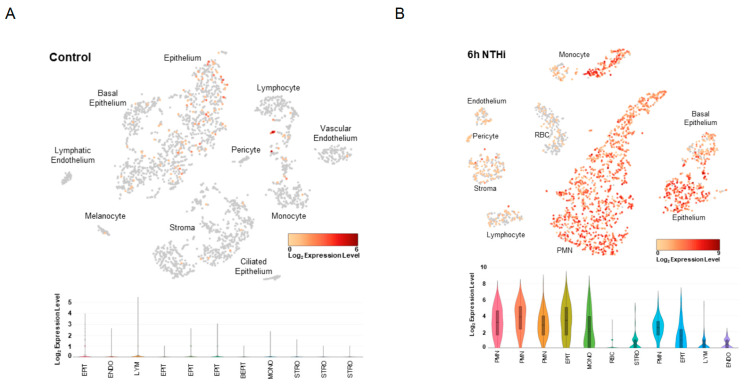
*Slpi* expression in wildtype murine middle ear prior to and at 6 h after NTHi inoculation. scRNA-seq expression profile of *Slpi* in (**A**) control murine ME in which some baseline expression can be observed in epithelial cells and some lymphocytes, and (**B**) 6 h post-NTHi inoculation in which *Slpi* is upregulated and expression can be observed in endothelium, stroma, lymphocytes, and red blood cells, with moderate expression in basal epithelium and strong expression in nearly all epithelial cells, polymorphonuclear cells, and a population of monocytes.

**Figure 4 ijms-26-01411-f004:**
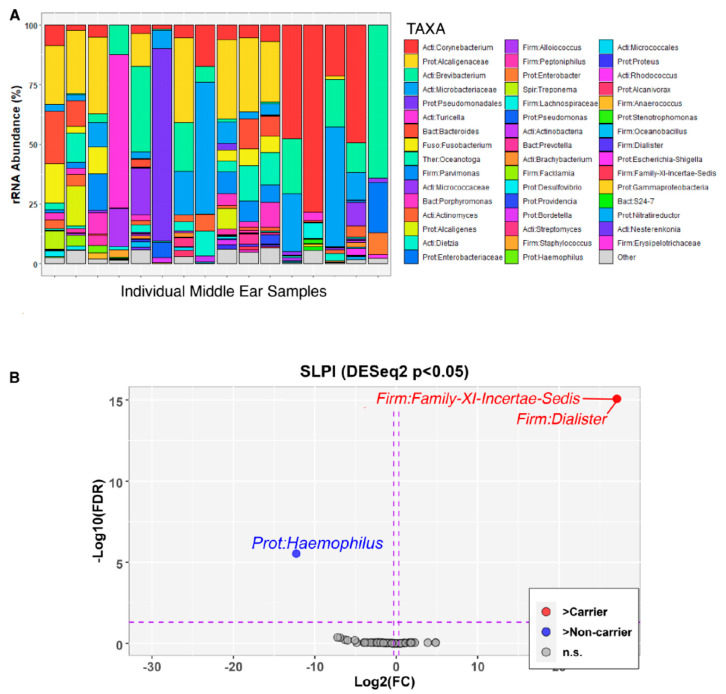
Relative abundance of individual taxa in the middle ears of carriers and non-carriers of the *SLPI* c.394+1G>T variant. (**A**) Individual profiles for middle ear microbiota. (**B**) DESeq2 analysis identified *Haemophilus* (FDR-adjusted-*p* = 2.97 × 10^−6^) as significantly enriched in non-carriers (*blue*) and *Family-XI-Incertae-Sedis* (FDR-adjusted-*p* = 8.38 × 10^−16^) and *Dialister* (FDR-adjusted-*p* = 8.38 × 10^−16^) significantly enriched in variant carriers (*red*).

## Data Availability

Variant information is available in the ClinVar database, accession number SCV004012948. Bulk mRNA-sequence data were previously deposited in dbGaP phs001941.v3.p1. Demultiplexed paired-end 16S rRNA sequence data were deposited in the NCBI Short Read Archive under accession number PRJNA439435. For access to scRNA-seq data from mouse middle ears, please email Prof. Allen F. Ryan: afryan@health.ucsd.edu.

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
