# Peer review of "A Novel SLPI Splice Variant Confers Susceptibility to Otitis Media in Humans"

_ijms, 2025, doi:10.3390/ijms26041411_

Round 1

Reviewer 1 Report

Comments and Suggestions for Authors

This manuscript investigates the relationship between the SLPI c.394+1G>T mutation and otitis media susceptibility, supported by a combination of genetic, microbial, and functional analyses. While this study provides valuable insights, several issues limit the robustness and generalizability of the findings. Below are specific concerns and suggestions for improvement.

1. Figure 1 in this manuscript overlaps significantly with Figure 1 in a previously published study (https://doi.org/10.1136/jmedgenet-2020-106844, https://pmc.ncbi.nlm.nih.gov/articles/PMC8218788/). Although the corresponding author authored the previous paper and this work could be part of a larger series, such reuse of images constitutes self-plagiarism and academic misconduct. It is recommended to rebuild Figure 1 with original content and ensure the proper citation of the previous study.

2. The sample size and diversity in this study are very limited. The saliva RNA analysis includes only 7 carriers and 9 non-carriers, while the microbiome study is similarly small. Additionally, the study population is drawn exclusively from a specific Filipino group, which restricts the generalizability of the findings. Expanding the sample size or tempering the aims of the project to reflect its exploratory nature would strengthen the manuscript. The authors should also explicitly acknowledge these limitations in the discussion section.

3. The study relies on saliva RNA-seq to demonstrate exon skipping of SLPI caused by the c.394+1G>T mutation. However, no direct data from middle ear tissues, the primary site of otitis media pathology, are presented. Since obtaining middle ear samples can be challenging, the authors could explore public single-cell databases, such as hECA (http://eca.xglab.tech/#/genePortrait?gene=SLPI), which provides cross-organ and cross-cell type comparisons. Incorporating such data would bolster the findings and provide indirect evidence for the mutation’s impact on middle ear tissues.

4. The study describes the generation and analysis of scRNA-seq data in detail but does not explicitly state whether these data are publicly accessible. Transparency is critical for reproducibility and further exploration by the scientific community. If the scRNA-seq data are available, the authors should provide repository links or accession numbers. If not, they should clarify how readers can request access.

5. The manuscript mentions the SpliceAI Δ score but does not provide sufficient context for interpretation. For instance, it is unclear whether a Δ score of 0.76 is high or low and what's the rationale behind this score. The authors should briefly explain the SpliceAI scoring system, including thresholds that help readers interpret its predictions.

6. The presentation of Figure 2 could be improved for clarity and informativeness. Figure 2A lacks individual-level microbiome data, which obscures sample variability. Providing per-patient microbiota percentages, in addition to the group-level summaries, would add value. Meanwhile, Figure 2B’s volcano plot is visually large but provides limited information. A concise table summarizing significant microbiota differences would be a more efficient and informative addition.

Author Response

This manuscript investigates the relationship between the SLPI c.394+1G>T mutation and otitis media susceptibility, supported by a combination of genetic, microbial, and functional analyses. While this study provides valuable insights, several issues limit the robustness and generalizability of the findings. Below are specific concerns and suggestions for improvement.

  1. Figure 1 in this manuscript overlaps significantly with Figure 1 in a previously published study (https://doi.org/10.1136/jmedgenet-2020-106844, https://pmc.ncbi.nlm.nih.gov/articles/PMC8218788/). Although the corresponding author authored the previous paper and this work could be part of a larger series, such reuse of images constitutes self-plagiarism and academic misconduct. It is recommended to rebuild Figure 1 with original content and ensure the proper citation of the previous study.

Authors have updated the language in the figure caption to reflect that this figure has been adapted from a version in a previous study. The last sentence of the Legend to Figure 1 that includes references to previous papers was edited as follows: “Adapted from Frank et. al 2021, this figure represents our most complete record of pedigree relations to date and was noted in previous publications for SPINK5, FUT2 and A2ML1 [3-6].” Authors would like to point out that the labelling schematic for variants and phenotype is different from the mentioned in publication, in which variants are denoted by ring colors and phenotype by fill color whereas in this manuscript phenotype is denoted by ring color and variant by quadrant fill color. Authors feel it is important to include previously identified genetic variants to highlight that there were individuals unresolved at the start of this study.

It should be noted that there are more individuals ascertained from the indigenous community (152) than those in the pedigree (139). Most of these additional individuals were ascertained at our last visit. However, despite multiple interviews with community members and general knowledge that they belong to the same tribe, we could not connect them reliably to the pedigree drawing, hence no new pedigree members were added to the 2021 version.

  1. The sample size and diversity in this study are very limited. The saliva RNA analysis includes only 7 carriers and 9 non-carriers, while the microbiome study is similarly small. Additionally, the study population is drawn exclusively from a specific Filipino group, which restricts the generalizability of the findings. Expanding the sample size or tempering the aims of the project to reflect its exploratory nature would strengthen the manuscript. The authors should also explicitly acknowledge these limitations in the discussion section.

 Authors recognize that sample size is a limitation of the study and have updated the discussion section, paragraph 6, with the addition of this last sentence: “Furthermore, sample size for the ME microbiome data is limited due in part to the samples themselves having low abundance of bacterial DNA by nature and, despite having collected more samples, a proportion of the samples often do not pass QC thresholds for read counts or quality of reads. Because these are community-based samples, we were not able to collect ME samples from individuals not actively experiencing OM, such as those indicated as “normal” or “healed” without eardrum perforations (Figure 1).” The RNA splicing study only used 3 samples as we felt that was sufficient to determine effect on splicing between wildtype, heterozygous and homozygous variant carriers.

For the RNA-seq analysis of saliva samples from Coloradan children, the following sentence was added to paragraph 2 of Methods section 4.5: “From post-hoc analysis using the RnaSeqSampleSize app (https://cqs-vumc.shinyapps.io/rnaseqsamplesizeweb/), given a sample size of n=30, maximum dispersion of 0.01 and 12,000 genes for testing, our RNA-seq analyses will have 98% power to detect an association.”

We agree that the findings are not readily generalizable to other populations because of the rarity of the SLPI variant and the lack of replication in other unrelated cohorts. However, our overall findings do support a role for SLPI as a candidate gene for human susceptibility to otitis media (please see the last sentence of the Discussion).

  1. The study relies on saliva RNA-seq to demonstrate exon skipping of SLPI caused by the c.394+1G>T mutation. However, no direct data from middle ear tissues, the primary site of otitis media pathology, are presented. Since obtaining middle ear samples can be challenging, the authors could explore public single-cell databases, such as hECA (http://eca.xglab.tech/#/genePortrait?gene=SLPI), which provides cross-organ and cross-cell type comparisons. Incorporating such data would bolster the findings and provide indirect evidence for the mutation’s impact on middle ear tissues.

Authors would like to highlight the following sentences 1-2 in paragraph 3 of Results section 2.1, in which we presented middle ear mucosa expression of SLPI in healthy individuals or those with chronic OM: “SLPI c.394+1G>T was also a prime candidate variant based on gene expression patterns: high RNA expression in ME mucosa from an OM cohort in Colorado (median TPM [transcripts per million] = 2863) [13]. In the Genotype-Tissue Expression (GTEx) database [14], SLPI RNA is predominantly expressed in the minor salivary gland (median TPM=13610), esophageal mucosa (median TPM=1418) and lung (median TPM=880); and in the Human Protein Atlas [15], high protein levels are reported in the nasopharynx and cervix (which contains mucosal epithelia) and medium levels in tissues that include the bronchus.” This paragraph has been updated to also include data from hECA, added as sentence 3: “In the human Ensemble Cell Atlas (hECA) [16], SLPI RNA has the highest expressing ratio in the esophagus (70.6% expressing), pleura (44.2%) and bronchi (44.1%) as well as perineural epithelial cells (73.3%), mesothelial cells (50.1%), intestinal gland cells (48.3%) and Paneth cells (48.3%). “

  1. The study describes the generation and analysis of scRNA-seq data in detail but does not explicitly state whether these data are publicly accessible. Transparency is critical for reproducibility and further exploration by the scientific community. If the scRNA-seq data are available, the authors should provide repository links or accession numbers. If not, they should clarify how readers can request access.

Added sentence to Data Availability for readers to contact afryan@health.ucsd.edu for access to scRNA-seq data.

  1. The manuscript mentions the SpliceAI Δ score but does not provide sufficient context for interpretation. For instance, it is unclear whether a Δ score of 0.76 is high or low and what's the rationale behind this score. The authors should briefly explain the SpliceAI scoring system, including thresholds that help readers interpret its predictions.

A brief explanation of SpliceAI and an overview of delta score interpretation has been added to Methods subsection 4.4, as paragraph 2.

  1. The presentation of Figure 2 could be improved for clarity and informativeness. Figure 2A lacks individual-level microbiome data, which obscures sample variability. Providing per-patient microbiota percentages, in addition to the group-level summaries, would add value. Meanwhile, Figure 2B’s volcano plot is visually large but provides limited information. A concise table summarizing significant microbiota differences would be a more efficient and informative addition.

Individual-level microbiota data were added to Figure 4A and Figure S6A. The volcano plots were improved as directed. Currently five figures in total pertain to the microbiota data.

Reviewer 2 Report

Comments and Suggestions for Authors

In this study, the authors identified a novel splice variant in the SLPI gene (c.394+1G>T) that co-segregates with otitis media in an indigenous Filipino cohort, demonstrating its impact on gene splicing, immune response, and microbiota composition in the middle ear and nasopharynx. These findings provide valuable insights into the genetic basis of otitis media susceptibility and emphasize SLPI's role in mucosal immunity and epithelial defense. However, the study does not explain how this truncated SLPI protein regulates the development of otitis media and the microbiota composition in the middle ear and nasopharynx. Nevertheless, the identification of this novel splice variant in the SLPI gene, which consistently co-segregates with otitis media, suggests that it could serve as a molecular marker for diagnosing individuals susceptible to otitis media.

Comments to improve the quality of the manuscript:

Figure 2: I suggest adding a PCA analysis comparing SLPI c.394+1G>T variant carriers (n=7) to non-carriers (n=9). Additionally, what are the ages of these samples (both SLPI variant carriers and non-carriers)? Age is a significant factor affecting microbiota composition, so this should be clarified.

Figure 3: Please label the forward (F) and reverse (R) primers and indicate their directions in panel 3A.

Figure 4: Do these two samples have similar sequencing depths and cell numbers? The authors should confirm that the low expression of SLPI in the control group is not due to lower sequencing depth or data quality. Additionally, the authors should perform clustering analysis and annotation (using either t-SNE or UMAP) for both samples simultaneously. Otherwise, the expression levels of SLPI may not be comparable across different samples.

Figure S4: There is a typo ("expressors")—this should be corrected. Also, why is the SLPI gene not included in the differentially expressed genes shown in Figure S4? In addition, some gene names are not fully displayed in the figure; these should be adjusted to ensure proper representation.

Results section 2.2-2.5: The authors could add more transitional sentences to explain the scientific questions they aim to address in each section. This would help clarify the purpose of the experiments presented in the current sections and how the findings in the results contribute to answering these questions.

I suggest moving the result sections 2.3 and 2.4 to the beginning of section 2.2.

Author Response

In this study, the authors identified a novel splice variant in the SLPI gene (c.394+1G>T) that co-segregates with otitis media in an indigenous Filipino cohort, demonstrating its impact on gene splicing, immune response, and microbiota composition in the middle ear and nasopharynx. These findings provide valuable insights into the genetic basis of otitis media susceptibility and emphasize SLPI's role in mucosal immunity and epithelial defense. However, the study does not explain how this truncated SLPI protein regulates the development of otitis media and the microbiota composition in the middle ear and nasopharynx. Nevertheless, the identification of this novel splice variant in the SLPI gene, which consistently co-segregates with otitis media, suggests that it could serve as a molecular marker for diagnosing individuals susceptible to otitis media.

Comments to improve the quality of the manuscript:

Figure 2: I suggest adding a PCA analysis comparing SLPI c.394+1G>T variant carriers (n=7) to non-carriers (n=9). Additionally, what are the ages of these samples (both SLPI variant carriers and non-carriers)? Age is a significant factor affecting microbiota composition, so this should be clarified.

Figure S4A-B in the supplement now include PCoA-Aitchison plots for the ME and NP microbiotas. Only the NP was nominally significant in these analyses, as noted in the legend for Figure S4C-D. Similarly, the legend to Figure S4D states this about the nasopharyngeal microbiota: “Beta-diversity between the two groups as measured by Aitchison distance is only nominally significant (nominal-p=0.04) after adjusting for sequencing batch (p=1x10-6) and age (p=0.67; see also Figure S2B).” This latter sentence was restated in Results section 2.5, paragraph 2, sentence 4. In the Materials and Methods, section 4.7, paragraph 2, sentence 6, it was mentioned that the Aitchison index was adjusted for age, sex, OM type and batch effects, however only sequencing batch was significant in the final model.

Figure 3: Please label the forward (F) and reverse (R) primers and indicate their directions in panel 3A.

“F” and “R” labels were added to revised Figure 2A.

Figure 4: Do these two samples have similar sequencing depths and cell numbers? The authors should confirm that the low expression of SLPI in the control group is not due to lower sequencing depth or data quality. Additionally, the authors should perform clustering analysis and annotation (using either t-SNE or UMAP) for both samples simultaneously. Otherwise, the expression levels of SLPI may not be comparable across different samples.

The following information was added to the Results section 2.4: “Prior to inoculation, the total number of cells expressing detectable Slpi mRNA was 129 out of 2,818 (4.6%). After NTHi inoculation, a marked increase in Slpi expression occurs across all cell types, with polymorphonuclear and epithelial cells demonstrating strong expression in the majority of each cell population. (Figure 3B). The number of cells expressing detectable Slpi mRNA at 6 hours after inoculation was 2,041 out of 2,557 (79.8%) cells in the sample. In the control sample, the highest level of expression was in epithelial cells, but the log2 median normalized expression was zero. In the 6-hour sample, the median normalized expression was 3.32-log2. As a control, we assessed expression of beta actin in the epithelial cells of each sample. Median Actb mRNA normalized expression in control epithelial cells was 3.24-log2, and for the infected sample was 3.85-log2.”

In the supplement, Table S3 was added to compare scRNA-seq metrics prior to and 6 hours after infection. This sentence was appended to Discussion, paragraph 3: “The dramatic infection-related increase in the number of cells expressing Slpi mRNA and in their expression levels, combined with the equivalent scRNA-seq metrics for the two samples (Table S3), indicate that the difference is not the result of technical issues.”

Figure S4: There is a typo ("expressors")—this should be corrected. Also, why is the SLPI gene not included in the differentially expressed genes shown in Figure S4?  

“Expressor” is also correct and more commonly used, though authors updated the caption to reflect the reviewer’s preference.

In addition, some gene names are not fully displayed in the figure; these should be adjusted to ensure proper representation.

All gene names are shown in Figure S1.

Results section 2.2-2.5: The authors could add more transitional sentences to explain the scientific questions they aim to address in each section. This would help clarify the purpose of the experiments presented in the current sections and how the findings in the results contribute to answering these questions.

Transitional sentences were added to the Results section as recommended by the reviewer.

I suggest moving the result sections 2.3 and 2.4 to the beginning of section 2.2.

Results and Methods sections were moved as prescribed.

Round 2

Reviewer 1 Report

Comments and Suggestions for Authors

The authors have answered my concerns. I've no additional questions.